# Risk and Resilience Variants in the Retinoic Acid Metabolic and Developmental Pathways Associated with Risk of FASD Outcomes

**DOI:** 10.3390/biom14050569

**Published:** 2024-05-10

**Authors:** Leo McKay, Berardino Petrelli, Molly Pind, James N. Reynolds, Richard F. Wintle, Albert E. Chudley, Britt Drögemöller, Abraham Fainsod, Stephen W. Scherer, Ana Hanlon-Dearman, Geoffrey G. Hicks

**Affiliations:** 1Department of Biochemistry & Medical Genetics, Max Rady College of Medicine, Rady Faculty of Health Sciences, University of Manitoba, Winnipeg, MB R3E 0J9, Canada; 2Department of Biomedical and Molecular Sciences, Queen’s University, Kingston, ON K7L 2V7, Canada; 3The Centre for Applied Genomics, The Hospital for Sick Children, Toronto, ON M5G 0A4, Canada; 4Department of Pediatrics and Child Health, Max Rady College of Medicine, Rady Faculty of Health Sciences, University of Manitoba, Winnipeg, MB R3A 1S1, Canada; 5Paul Albrechtsen Research Institute CancerCare Manitoba, Winnipeg, MB R3E 0V9, Canada; 6Children’s Hospital Research Institute of Manitoba, Winnipeg, MB R3E 3P4, Canada; 7Centre on Aging, University of Manitoba, Winnipeg, MB R3T 2N2, Canada; 8Department of Developmental Biology and Cancer Research, Institute for Medical Research Israel-Canada, Faculty of Medicine, The Hebrew University of Jerusalem, P.O. Box 12271, Jerusalem 9112102, Israel; 9Department of Molecular Genetics and McLaughlin Centre, University of Toronto, Toronto, ON M5G 1L7, Canada; 10Program in Genetics and Genome Biology, The Hospital for Sick Children, Toronto, ON M5G 0A4, Canada

**Keywords:** fetal alcohol spectrum disorder, FASD, retinoic acid, genetic variants, FASD risk outcomes, prenatal alcohol exposure, PAE, rare neurodevelopmental disorders, whole exome sequencing, biomarkers

## Abstract

Fetal Alcohol Spectrum Disorder (FASD) is a common neurodevelopmental disorder that affects an estimated 2–5% of North Americans. FASD is induced by prenatal alcohol exposure (PAE) during pregnancy and while there is a clear genetic contribution, few genetic factors are currently identified or understood. In this study, using a candidate gene approach, we performed a genetic variant analysis of retinoic acid (RA) metabolic and developmental signaling pathway genes on whole exome sequencing data of 23 FASD-diagnosed individuals. We found risk and resilience alleles in *ADH* and *ALDH* genes known to normally be involved in alcohol detoxification at the expense of RA production, causing RA deficiency, following PAE. Risk and resilience variants were also identified in RA-regulated developmental pathway genes, especially in *SHH* and *WNT* pathways. Notably, we also identified significant variants in the causative genes of rare neurodevelopmental disorders sharing comorbidities with FASD, including *STRA6* (Matthew–Wood), *SOX9* (Campomelic Dysplasia), *FDG1* (Aarskog), and 22q11.2 deletion syndrome (*TBX1*). Although this is a small exploratory study, the findings support PAE-induced RA deficiency as a major etiology underlying FASD and suggest risk and resilience variants may be suitable biomarkers to determine the risk of FASD outcomes following PAE.

## 1. Introduction

Fetal Alcohol Spectrum Disorder (FASD) is a common neurodevelopmental disorder that affects an estimated 2–5% of the population [1]. Individuals across the spectrum show behavioural, adaptive, and cognitive abnormalities, which can be as serious in those without any physical features as in those with the most severe form of FASD, Fetal Alcohol Syndrome (FAS) [2]. Diagnosing individuals with FASD remains difficult as most do not present with FASD sentinel facial features [3], and it requires assessments involving many medical disciplines. In the absence of FASD physical hallmarks, confirmation of PAE and a psychological profile that meets diagnostic guideline criteria are necessary for diagnosis [2]. As such, additional diagnostic tools are needed to identify FASD for early intervention and better outcomes. Due to the complex nature of the disorder, FASD has many contributing risk factors including alcohol dosage [4,5], duration of exposure and gestational timing [6,7,8,9], maternal nutrition, metabolism, socioeconomic factors, stress, and genetic variant composition [10,11,12,13,14].

Genetic composition is a well-established risk factor in FASD. PAE animal models have shown that different strains of the same animal species have different susceptibilities to PAE [15]. Human twin studies on individuals with FASD have shown that monozygotic twins share 100% concordance for their FASD diagnoses, while fraternal twins have around 56% and non-twin siblings only have 46% [16]. Importantly, the majority of individuals with PAE do not develop FASD [17], suggesting that there is a genetic vulnerability that sensitizes an individual to develop FASD in the presence of PAE. One can hypothesize that allelic variants found in such genes can act as risk or resilience factors that could either drive or protect against FASD outcomes.

Research in PAE animal models has revealed many potential gene targets for investigation within PAE-impacted pathways, such as Shh and Wnt, that could host FASD risk variants [15]. Retinoic acid (RA) is a developmental signaling pathway shown to regulate many PAE-impacted pathways [18]. Moreover, Vitamin A (retinol) supplementation rescues PAE phenotypes in animal studies, while RA-deficiency phenocopies many of the craniofacial and neurological malformations found in PAE animal models and individuals with FASD [18,19,20,21]. Enzymes involved in RA biosynthesis have been shown to participate in ethanol detoxification at the expense of RA production. These findings indicate that variants in RA metabolic genes are likely candidates in determining whether an individual prenatally exposed to alcohol will develop FASD [18,21].

Many rare neurodevelopmental disorders (NDDs), such as CHARGE and Smith–Lemli–Opitz syndrome (SLOS), phenocopy FASD in terms of their specific craniofacial, neurodevelopmental, or behavioural manifestations [20]. Interestingly, many of the causative genes of these rare NDDs are regulated by RA signaling and crosstalk with well-established PAE targets such as SHH and WNT. Therefore, it is likely that allelic variants within the causative genes of these NDDs may also sensitize individuals to develop FASD following PAE, given their shared co-morbidities and signaling pathways. Haploinsufficiency in causative genes may be enough to disrupt the signaling pathway [15] and produce a similar phenotype seen in both NDDs and FASD.

Taken together, RA serves as a master regulator of this signaling network, which encompasses well-known PAE-implicated pathways and the causative genes of rare NDDs directing embryonic development [18,20]. Variants found within the RA metabolic and signaling network may sensitize or protect individuals against developing FASD following PAE. Furthermore, such variants may serve as genetic biomarkers of risk or resilience of FASD outcomes following PAE. The goal of this exploratory study was to investigate the variants found in the genes comprising the RA signaling network in 23 individuals with FASD and to discover novel FASD risk and resilience alleles to further our understanding of this disorder. To examine the variants found in our FASD cohort, the RA signaling network was divided into the following three candidate gene lists: (1) RA and alcohol (ethanol) metabolism genes, (2) RA-regulated developmental pathway genes (such as SHH and WNT), and (3) the causative genes of rare NDDs and their direct signaling targets.

## 2. Methods

### 2.1. Participants

The 23 children in this study are part of the Canadian NeuroDevNet/Kids Brain Health Network FASD Study Cohort [22,23]. Participants were diagnosed at the Manitoba FASD Centre where the assessment and diagnosis of FASD are based on an experienced multidisciplinary team approach including medical evaluation of developmental pediatricians (MD) and medical geneticists (MD) with dysmorphology training using the revised Canadian FASD Diagnostic Guidelines [2]. These guidelines incorporate the use of the Washington four-digit code. Differential diagnosis carefully considers other genetic, neurodevelopmental, medical, and environmental contributors to an individual presentation. The dedicated diagnostic team also includes qualified and certified speech and language pathologists, Clinical Psychologists (PhD), social workers (MSW), and clinic coordinators. Relevant birth records were obtained, and a comprehensive review of prenatal alcohol and substance exposure history, medical record review, and complete social history was conducted as part of the assessment for FASD-related diagnoses. Eleven children were male and 12 were female. In our FASD cohort, there were 3 pairs of siblings, each pair had matched diagnoses, 2 pairs were both diagnosed with partial FAS (pFAS), and the other pair with ARND. Ten children were of European ancestry, and the remaining 13 were of Indigenous ancestry, either First Nations (*n* = 10), Inuit (*n* = 1), or Metis (*n* = 2). Six individuals were diagnosed with FAS, 12 with pFAS, and 5 with alcohol-related neurodevelopmental disorder (ARND). Research ethics board approvals were obtained at collection centres and either written or verbal assent, as well as written consent, was given by all participants and their respective caregivers or legal guardians, respectively. Saliva samples were then obtained from the children; the details of sample collection and storage can be found elsewhere [23]. Approval of this study was granted by the Health Research Ethics Board at the University of Manitoba, (REB Study Number: H2010:206).

### 2.2. Whole Exome Sequencing

DNA was extracted from the 23 participants’ saliva samples and sent for whole exome sequencing at the Center of Applied Genomics located at The Hospital for Sick Children in Toronto, Canada. DNA samples were first ligated with SOLiD A1 and P1 adaptors to undergo PCR amplification (Thermo Fisher Scientific, Waltham, MA, USA). Exome capture was then completed according to the Agilent SureSelect Protocol (Agilent Technologies, Inc., Santa Clara, CA, USA). Upon completion, the samples underwent a PCR amplification to apply barcodes for multiplex sequencing. The captured library was then purified using AMPure XP beads following the Agilent SureSelect Protocol. Paired-end sequencing was completed with ABI SOLiD 5500xl (Thermo Fisher Scientific, Waltham, MA, USA) on the purified library and then analyzed using the Applied Biosystems corona pipeline to generate sequencing data. These paired end-reads generated by the ABI SOLiD were mapped to the human reference genome (hg19) using BFAST [24] for 75 bp forward reads and BFAST implemented BWA version 0.6.5a [25] for 35 bp reverse reads. MarkDuplicates was used to remove any duplicate paired-end reads that were found [26]. The duplicate-free alignments were then refined using local realignment with SRMA version 0.1.15 [27]. GATK version 1.1.28 was then used for variant calling [28] (for QC see Section 2.4 below). The resulting variant analysis had a sequencing depth of 30×.

### 2.3. Candidate Gene Lists

All candidate gene lists found in this paper were manually curated based on a review of literature sources, excluding papers studying cancer. Literature sources for the RA and alcohol metabolism candidate gene list were compiled based on the genes that encompass the RA and alcohol metabolism pathways, as well as a selection of variants previously associated with alcohol consumption, dependence, and clearance (Appendix A). To assemble the list of candidate developmental genes found in RA-regulated PAE-impacted pathways, the literature was searched for known RA-controlled genes active during development affected by PAE, as well as their direct interaction partners (Appendix A, Appendix A). Literature sources examining the causative genes of rare NDDs with shared specific craniofacial, neurological, as well as behavioural co-morbidities with FASD and their directly interacting genes were used to compile the third candidate gene list (Appendix A), as discussed in our recent review [20]. A fourth candidate gene list was also compiled that includes synthesizing enzymes and receptors for neurotransmitters, forebrain development genes, and collagen genes (Appendix A; the results are not discussed in this report but can be found in Appendix A).

### 2.4. Variant Analysis of Whole Exome Sequencing Data Using Candidate Gene Approaches

Aligned variants found in the whole exome sequencing data of the FASD cohort were imported into R, and all variants found in the genes from our gene lists of interest were compiled; the frequencies of variant alleles in our FASD cohort were then determined. The corresponding 1000 Genomes Project global allelic frequencies were used as controls (NCBI) (Figure 1). Variants further analyzed included those found in exons, introns, 3′ UTR, 5′ UTR, and upstream regions. As the smallest detected frequency in our cohort was 1/46 (2.17%), a minor allele frequency (MAF) threshold filter of ≥2.17% (QC filtering parameters) was applied to all alleles in the 1000 Genomes Project dataset for variants presented in the results tables. The allele frequencies for the FASD cohort and the 1000 Genomes Project control dataset were then compared to each other using a chi-squared test. A Benjamini–Hochberg (BH) multiple test correction with a 0.05 significance threshold was then applied to the chi-square test *p*-values (number of tests (*n*) = 1100) with a resulting *p*-value threshold of 0.024. To account for variants with known alcohol associations, such as alcohol consumption and dependence that may have low frequencies, as well as any known rare pathogenic variants that may be found within developmental genes in our FASD cohort, known variants with frequencies below 2.17% in the 1000 Genomes Project dataset were included in the total number of tests. All statistically significant genetic variants were then sequentially examined in the literature and annotated with PolyPhen2 and SIFT scores [29,30] based on their impact on gene function and association to clinically relevant phenotypes to determine risk and resilience variants (Figure 1). Discovered risk and resilience variants were then annotated with CADD scores [31]. For predictive tools, we considered deleterious SIFT and possibly damaging or damaging PolyPhen2 predictions, as well as CADD scores > 10, to be potentially damaging to gene function [29,30,31]. It should be noted that the majority of these presumptive risk and resilience alleles require further experimentation to confirm their true impact on the severity of PAE teratogenesis.

### 2.5. TaqMan SNP Genotyping Assays

Following variant analysis of the FASD whole exome sequencing data, 16 genetic variants of interest were selected for validation by TaqMan probes ordered from Applied Biosystems (Thermo Fisher Scientific, Waltham, MA, USA). As the RA and alcohol metabolism pathways are the primary focus of this paper, variants in these pathways were prioritized if they were presumed pathogenic variants and/or planned to be assessed in future *Xenopus* and mouse model studies. TaqMan genotyping assays were conducted following the manufacturer’s guidelines. TaqMan reactions were performed in a 20 μL reaction volume in a Bio-Rad Hard-Shell 96-well plate and run using a Bio-Rad CFX96 Real-Time PCR Detection System (Bio-Rad, Hercules, CA, USA). The TaqMan genotyping results were then analyzed using Bio-Rad CFX Maestro Software (version 4.1.2433.1219) (Bio-Rad, Hercules, CA, USA) to determine the genotypes for each individual in the FASD cohort. All but one probe (rs7978237, *NCOR2*) amplified sufficiently to validate the WES genotype in all individuals.

### 2.6. Determining Differences between FASD Diagnostic Groups and Pathway Analysis

Statistically significant variants were grouped as follows: those unique to individuals with FAS, pFAS, or ARND and those significant variants that were common to both FAS and pFAS individuals (the two diagnostic categories that share clinical sentinel facial features) (see Appendix A). For genetic variants shared by both FAS and pFAS, the number of risk alleles in a given gene per person was determined, and this number was included in the represented major PAE-implicated pathway (RA, SHH, WNT, FGF, TGF- β, RAS-MAPK, RHO-RAC, and mTOR). The following operations were then applied: (1) If one variant in a gene was common to pFAS and FAS, other variants within that common gene were included. (2) Variants found in common to all three diagnostic groups were not included. (3) If a gene was found to not interact with any of the developmental pathways, then variants within that common gene were also not considered. The total number of risk alleles per pathway was compiled and then plotted using principal component analysis (PCA) using R software (version 4.1.2) to determine the degree of separation between FAS and pFAS diagnostic groups.

## 3. Results

### 3.1. The FASD-Diagnosed Cohort Has Increased Frequencies of RA and Alcohol Metabolism Gene Variants Known to Be Associated with FASD and Alcoholism

Alcohol dehydrogenases (ADHs) and acetaldehyde dehydrogenases (ALDHs) comprise a two-step sequential oxidation pathway where these enzyme families convert ethanol (alcohol) to acetic acid. Some members of these families also perform the two sequential oxidations of retinol (vitamin A) to produce RA (Figure 2). Many variants of these two enzyme families have been associated with increased alcohol consumption, dependence, and altered enzyme activity, but only a few have thus far been associated with FASD [32]. Our group and others have shown in animal models that alcohol-induced RA deficiency during gastrulation can result in most known FASD developmental phenotypes [33,34]. Accordingly, certain genetic variants of RA metabolism may predispose to RA deficiency following PAE, and these same variants would be found with increased allele frequencies in children diagnosed with FASD in this cohort. Using a candidate gene approach, all detected variants in the 44 genes of RA metabolism were analyzed (Figure 2 and Appendix A) in the FASD cohort whole exome sequencing data.

As hypothesized, significant variants were found in the genes coding for alcohol metabolic enzymes and the broader RA metabolic pathway associated with FASD. Specifically, 15 variants were more frequently observed in individuals with FASD, while five alleles were less frequently observed in individuals with FASD when compared to the 1000 Genomes Project dataset (Table 1); moreover, over half of these variants were previously shown to be associated with increased alcohol consumption, dependence, or altered enzyme activity (Table 1 and Figure 2; known alcohol risk variants enriched in our study are denoted by a red dot and resilience variants by a green dot). Known FASD protective variants in *ADH1B* were found to be significantly reduced in our FASD cohort (rs1229984 and rs2006702) [32,35,36]. These two variants modify the kinetic speed and binding efficiency of *ADH1B*, changing how fast alcohol is oxidized and, consequently, are associated with alcohol consumption and dependency. Noteworthy variants also associated with alcohol dependence and consumption found to be enriched in the cohort included the risk alleles rs1614972 and rs698 in *ADH1C*, along with rs3762894 and rs671 in *ADH4* and *ALDH2*, respectively [37,38,39,40].

Other significant variants in other RA metabolism genes that could impact the severity of alcohol teratogenicity in a PAE individual were also found (Figure 1 and Table 1). One risk variant in *RXRG*, rs2134095, which affects the splicing of *RXRG* [41] was found to be enriched in the FASD cohort. *RXRG* is necessary to mediate RA-regulated transcription; therefore, rs2134095 may potentially hinder RA signaling efficiency in impacted individuals. Two risk variants, rs971756 and rs11857410, were found in *STRA6,* which is an essential transporter of retinol into the cell [42]. Interestingly, both *STRA6* variants have been associated with Matthew–Wood syndrome, a genetic syndrome that has many comorbidities of FASD such as craniofacial, ocular, and cardiac defects. Both rs971756 and rs11857410 have a CADD score of over 10, making them likely damaging variants that reduce the amount of retinol entering the cell to be converted to RA, thereby reducing RA signaling, and likely sensitizing a PAE individual to develop FASD. New potential FASD risk variants in RA metabolism with damaging PolyPhen, SIFT, and/or CADD scores were also found in *BCO2*, *CYP26C1*, *DGAT1*, *FABP4*, *LIPC*, and *NCOR2*; these variants could influence RA levels and signaling, potentially increasing the risk that a PAE individual will develop FASD.

**Figure 2 biomolecules-14-00569-f002:**
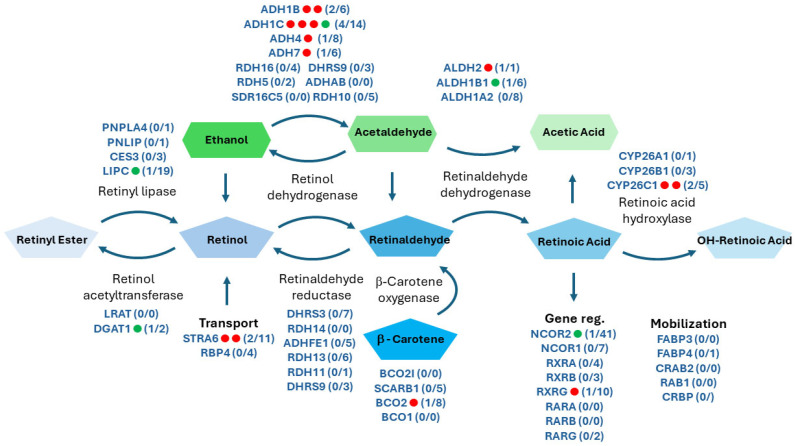
The FASD cohort is enriched in FASD risk alleles throughout the retinoic acid and alcohol (ethanol metabolic pathways and deficient in resilience alleles). Ethanol (green highlight) and retinoic acid (blue highlight) metabolic pathways are shown. All major participating genes involved in specific metabolic steps are indicated across their respective enzymatic reaction. Red and green dots listed to the side of gene names represent enriched FASD risk and resilience alleles found in that gene, respectively, that were found to be significant in the FASD cohort when compared to the allele frequencies cited in the 1000 Genomes Project dataset. The numbers next to gene names show the number of risk and resilience alleles over the total number of detected variants for that gene that passed our threshold assessment (Figure adapted from Parihar M. et al., 2021 [43]).

### 3.2. The FASD Cohort Is Enriched in Both Risk and Resilience Variants in Retinoic Acid-Regulated Developmental Pathways

Research in animal models of PAE has shown that many important developmental pathways are impacted by alcohol including SHH, WNT, and TGF-β among others [15,20]. Many of these pathways are either directly or indirectly regulated by RA signaling during development [18,20]. As such, genes within PAE-impacted developmental pathways that are known to be regulated by RA were chosen to be examined (Appendix A). Interestingly, more significant frequency changes in resilience variants (16 reduced) were detected, compared to frequency changes in risk variants (11 enhanced).

When examining variants found in the genes in RA-controlled developmental pathways, 66% (18/24) were found to affect craniofacial, heart, and/or bone development. Eight significant variants associated with craniofacial malformations such as non-syndromic cleft lip and palate, mandibular prognathism, ocular abnormalities, and tooth defects are found in *AXIN2*, *BMP2*, *BMP4*, *NOTCH3*, *PTCH1*, *SOX10*, and *WNT10A* (Table 2 and Appendix A) [44,45,46,47,48,49,50,51]. The variants in these genes could increase the risk of developing FASD in a PAE individual given their potential to cause similar facial features as seen in children with FASD. Six significant variants within *AXIN1*, *AXIN2*, *CAV1*, *GLI1*, RYR2, and *SFRP4* genes were discovered that have an impact on heart development and function [52,53,54,55,56], which could worsen cardiac malformations in FASD individuals. Four significant variants in the *BMP2*, *WNT10B,* and *WNT16* genes were found and could have an impact on bone mineral density, ossification defects, or increased risk of bone fracture [57,58,59,60].

Additional significant variants were discovered in RA-regulated pathways that impact different aspects of development or tissue function that could adversely affect FASD individuals (Table 2 and Appendix A). Two variants with damaging PolyPhen and SIFT predictions were found in genes of the SHH pathway: rs2592595, associated with ectrodactyly, and rs11573590, found in *GLI2* and *PTCH3,* respectively [61]. One significant variant was found in the WNT pathway, rs34072914 in *WNT9B*. This variant is associated with Mayer–Rokitansky–Küster–Hauser syndrome [62] and has a CADD score of 12.75, making it potentially pathogenic. Three significant variants were also detected in *NOTCH* genes: rs35769976 in *NOTCH3*, which has been found to lead to a disruption of its protein structure [63], and rs1048672 and rs8192591 in *NOTCH3* and *NOTCH4*, respectively, which were both found to have damaging PolyPhen and SIFT predictions. rs1140475 in *EGFR* is another significant variant that is located in the protein kinase domain and associated with temporomandibular disorder [64,65].

### 3.3. The FASD Cohort Is Enriched in Risk Variants in the Causative Genes of Rare Neurodevelopmental Disorders

Many rare neurodevelopmental disorders (NDDs), such as 22q11.2 deletion syndrome (22q11.2DS) and Smith–Lemli–Opitz syndrome (SLOS), share FASD comorbidities and physical and neurological features in addition to their behavioural outcomes [20]. Given the comorbidity between FASD and these rare NDDs, we wondered whether certain variants in these rare disease genes could sensitize FASD risk outcomes in children with PAE. Accordingly, variants found in the FASD cohort within the causative genes of NDDs and their downstream gene targets were investigated (Appendix A). The FASD cohort was found to be statistically significant in 15 risk and 5 resilience alleles when compared to the 1000 Genomes Project dataset (Table 3 and Appendix A).

Several potential FASD significant variants in the causative genes and/or their interaction partners associated with Campomelic Dysplasia, Apert, SLOS, 22q11.2DS, Three M Syndrome 2, and Aarskog syndromes were detected in our FASD cohort (Table 3 and Appendix A). Eleven statistically significant variants with the potential to play a role in the development of FASD in genes found in the 22q11.2 region were discovered in our FASD cohort. Notable among these were two significant variants found in *TBX1*, rs5746826, and rs41298840, the most well-characterized gene in 22q11.2DS. rs55975541 in *CDC42BPG*, a variant with damaging PolyPhen and SIFT predictions, was found to be enriched; CDC42BPG is downstream of CDC42, the direct downstream target of the causative gene of Aarskog syndrome, *FGD1* [20]. The T variant of rs2229989 in *SOX9*, the causative gene of Campomelic Dysplasia [66], was also found to be more prevalent in our FASD cohort when compared to controls. The A and G alleles of rs2229989 result in an H169Q substitution that has only 46% of the activity of the WT enzyme and was found in a patient with mild Campomelic Dysplasia [66]. While the T allele results in a synonymous mutation, it did have a CADD score of 11.83, making it potentially pathogenic [15,21]. When examining variants within the *FGF* and *FGFR* genes, the FASD cohort was found to be enriched in the A allele of rs147057 in *FGFR2*, which has been found in a patient with Apert syndrome [67]. Cholesterol metabolism genes were also examined as SLOS—another NDD with shared FASD comorbidities—characterized by cholesterol deficiency due to mutations in the cholesterol-synthesizing enzyme DHCR7 [68]. A significant allele, rs2230808 in *ABCA1*, which plays an important role in cholesterol homeostasis by its function in the reverse cholesterol transport pathway, was detected. The FASD cohort was found to be enriched in the C allele of rs2230808, which has been associated with more severe phenotypes in SLOS patients [69]. Lastly, we found a significant allele, rs10932816, in *OBSL1*, one of the causative genes of Three M Syndrome 2 [70] that has a damaging SIFT prediction.

### 3.4. Determining Differences between FASD Diagnosis and Pathway Polygenic Risk Score Analysis

There is extensive evidence in human and animal studies that there is a clear underlying genetic component to FASD. Therefore, it should be possible to separate individuals with a FAS and pFAS diagnosis by their genetic differences in PCA. Additionally, more severe FASD clinical outcomes, including sharing sentinel facial features, may have increased genetic variants in the developmental pathways known to regulate the phenotypes observed. As an explorative exercise to test the hypothesis of genetic variants sorting by diagnosis, all candidate genetic variants found in common within both FAS and pFAS diagnostic groups were identified and then grouped by gene (Appendix A). The genes were then grouped by their interactions with developmental pathways—RA, SHH, WNT, FGF, TGF-β, RAS-MAPK, RHO-RAC, and mTOR [15,18,20] (eight pathways). Perhaps not surprisingly, a global PCA analysis of all genetic variants detected in the FASD cohort identified ethnicity as the largest principal component (Appendix A); however, PCA analysis of the significant variants in this study did not (Appendix A). Surprisingly, PCA analysis of genetic variants significantly enriched in the aforementioned developmental pathways sorted by diagnosis and, furthermore, with little bias towards ethnicity (Figure 3). RA was the most highly represented pathway with 44 significant variants (14%), followed by SHH at 29 variants (9%), WNT at 22 variants (7%), TGF-β at 18 variants (6%), RHO/RAC and mTOR at 13 variants (4%), and RAS/MAPK and FGF each at 11 variants (4%). Interestingly, 44% of the genes found to be common or unique to FAS and pFAS groups are causative genes of NDDs such as CHARGE, SLOS, and 22q11.2DS. Potentially pathogenic variants were discovered in the FAS and pFAS groups, including those known to cause craniofacial malformations, such as *SOX9* (Campomelic Dysplasia), *STRA6* (Matthew–Wood), *CHD7* (CHARGE), *MID1* and *MED15* (Opitz–Kaveggia syndrome), and several 22q11.2DS genes [20]. Unexpectedly, two individuals were found to be heterozygous for a known pathogenic variant, rs138659167, in the *DCHR7* gene for SLOS [71]. pFAS had the largest number of unique genes at 26, FAS had nine, and the ARND group had one (Appendix A).

## 4. Discussion

Herein we report 15 new risk variants and five new resilience variants in RA metabolism that have significant allele frequency changes in a cohort of 23 children diagnosed with FASD. Many impacted genes were found to have multiple associated risk or resilience variants that may independently affect gene and protein function. Allele frequencies of two *ADH1B* variants previously associated with FASD, rs2066702 and rs1229984, which are both protective variants associated with increased alcohol oxidation and less consumption, were found to be reduced [32,35,36]. Finding a significant reduction in the protective alleles of both rs2066702 and rs1229984 supports the employed candidate gene approach and the methodology used in this study. Moreover, eight new FASD-associated variants were found in alcohol metabolism genes [37,38,39,40,72,73,74,75]. Six of the eight alcohol metabolism variants are risk alleles; the other two variants are resilience alleles (Table 1 and Figure 2). Outside of the shared alcohol oxidizing enzymes, we identified nine additional new FASD variants in the remaining RA metabolism genes (seven risk and two resilience variants).

It is important to note that RA variants previously found associated with alcohol dependency and addiction are not functionally associated. These risks are not directly relevant to FASD outcomes in the embryo itself; rather, they are associated with the risk of embryo/developmental outcomes within the context of our hypothesis that PAE results in an RA deficiency within the embryo proper. Under conditions of PAE, the 15 new risk variants may enzymatically sensitize individuals to poorer maternal alcohol detoxification and, in turn, metabolically result in a state of RA deficiency in the embryo. When undergoing PAE, the accumulating acetaldehyde acts as a toxin in the embryo and its elimination is thus the priority [76,77]. As the alcohol and RA metabolic pathways share familial enzymes, both retinol and retinaldehyde dehydrogenases will help to detoxify alcohol and acetaldehyde during PAE [33,76,77]. Presumably, this is why many of the shared alcohol metabolism variants were originally associated with alcohol behaviour risks in the mother and father [32], and it would be interesting to determine which are inherited maternally or paternally. Here, their direct role in alcohol detoxification can explain enrichment in frequencies in a high number of risk alleles in these alcohol metabolism genes. Usurping retinol and retinaldehyde dehydrogenase for alcohol detoxification also comes at the cost of reduced synthesis of RA, particularly when it is essential to guide cellular spatiotemporal patterning [33]. Most individuals who are prenatally exposed to alcohol do not develop FASD [17], which is likely due to the robustness of the RA signaling network being able to adjust to acute PAE accordingly [43]. However, when damaging variants are within the RA and alcohol metabolism genes and already perturb the RA pathway, PAE as an additional factor might disrupt the pathway so severely that FASD develops. Similarly, the reduced frequency of five resilience alleles in our FASD cohort also favours alcohol detoxification at the expense of RA production, leading to RA deficiency.

When viewed through this lens, the FASD cohort appears to be particularly sensitized to alcohol teratogenesis, as individuals harbour multiple damaging variants throughout the shared RA and alcohol metabolism pathways. The rs971756 and rs11857410 variants in *STRA6*, which are associated with Matthew–Wood syndrome and were enriched in the FASD cohort, are a good example of this [42]. These *STRA6* variants were also found uniquely in pFAS and FAS individuals, the more severe forms of FASD, demonstrating the importance of RA signaling when determining an individual’s risk of FASD outcomes following PAE.

When taken together, these findings support the proposed hypothesis, wherein PAE-induced RA deficiency is a potential underlying etiology of FASD, particularly during early gastrulation. Animal models have supported this hypothesis, both in studies conducted by our group and others, and now here, with the enrichment of risk alleles of RA metabolism in a human FASD cohort [33,34]. Our FASD cohort is small, and while it supports the hypothesis, these results need to be validated in a larger multiethnic cohort. To our surprise, there were not many damaging variants found in the *ALDH1A2* gene (Appendix A), which has been shown to be competitively inhibited by ethanol during embryogenesis, thereby causing PAE-induced RA deficiency [33]. It may be that damaging variants in the *ALDH1A2* gene may result in loss of pregnancy following PAE, as observed in animal models targeting the *ALDH1A2* gene [78]. It is well known that PAE is associated with higher numbers of stillbirths and miscarriages [79]. FASD cohort studies that include parent genotyping, and dosage and duration of PAE insult will be able to explore this avenue of research.

Extending the candidate gene approach analysis to RA-interacting major developmental pathways known to be impacted by PAE, such as SHH, WNT, and TGF-β, identified many new potential FASD risk and resilience alleles (Table 2 and Appendix A) [15,20]. Interestingly, a greater number of resilience than risk alleles, 16 vs 11, were found in these pathways, compared to those found in the RA metabolism pathway. This was a surprising result given that animal models have shown these pathways are directly impacted by PAE, and when perturbed can exacerbate PAE teratogenesis [80,81]. Although these PAE-impacted pathways can be viewed as independent of each other, they are co-dependent and function through defined signaling gradients [82]. If one is disrupted, the others are consequently perturbed, perhaps amplifying disruptive effects following PAE and/or RA deficiency.

This study identified 15 risk and five resilience variants in causative genes of rare NDDs that share clinical comorbidity with FASD. Our group has previously identified several rare NDDs that share specific comorbidities with children with FASD, including the causative genes and/or their interaction partners in Campomelic Dysplasia, Apert, SLOS, 22q11.2DS, Three M Syndrome 2, and Aarskog syndromes (Table 3) [20]. Rare NDDs that share many co-morbidities with FASD may offer unique insights into the underlying genetics of FASD. Most of these NDDs are rare disorders requiring recessive homozygosity within a single gene. As such, these causative variants are likely to highlight a single genetic developmental pathway that plays a key role in the development of NDD phenotypes. Therefore, genetic variants may exist in the shared NDD gene pathway that could be sensitized following acute PAE and result in similar clinical NDD comorbidities. Finding 20 new genetic variants in NDD rare disease genes with the potential to influence PAE teratogenesis provides initial evidence that this may be the case. It is also important to highlight that we also found many low-frequency alleles that did not pass the filtering threshold in some known causative genes of NDDs, *CHD7*, *BRD4*, and a known SLOS pathogenic variant in *DHCR7* [20,71], which could be considered FASD risk alleles. The SLOS pathogenic variant requires recessive homozygosity to develop the disorder; therefore, these children were not misdiagnosed, but are rather potentially sensitized to PAE by being heterozygous for this variant. If these findings hold true in larger cohort studies, this could increase the number of candidate genetic biomarkers for PAE teratogenesis, and the number of new targets of potential therapeutics for both NDD and FASD children.

Another unexpected finding was that newly identified risk variants are able to separate FAS and pFAS individuals within the FASD cohort by assessing variants found in genes shared between these two groups. However, this was only possible when we categorized shared variants using pathway-based polygenic risk score analysis, which is a higher-level approach that shows the impact of variants on major PAE-impacted signaling pathways (such as RA, SHH, and WNT). This assessment also required that we account for low-frequency alleles present in the cohort that were found in the shared genes of FAS and pFAS individuals (Appendix A). However, it is interesting to speculate that the different clinical diagnoses may align with high principal components associated with major developmental pathways. This may be due to an underlying connection between all of these pathways, as they are all involved in the regulation of neural crest cell development [20]. FASD and NDDs share co-morbidities and could be considered neurocristopathies due to their defects in neural crest cell lineage development that explain many of the shared craniofacial, neurological, and behavioural deficits seen in these disorders. Variants that identify risk alleles associated developmentally with FASD sentinel facial features may not greatly increase the diagnosis of FAS and pFAS in children with PAE. However, these results need to be validated in larger multiethnic FASD studies.

It is important to recognize this exploratory study has limitations. First, due to the small study cohort size, these potential genetic biomarkers need to be validated in a larger multiethnic and multi-diagnosis FASD cohort. Second, the discovered variants in this paper are only probable FASD risk and resilience alleles and require additional independent replication and validation to confirm their impact on protecting and sensitizing individuals to FASD. Third, the small sample size and the ability to segregate the FASD diagnosis groups from one another using pathway polygenic risk scores based on shared genetic variants found in the individuals with FAS and pFAS must also be replicated in larger multiethnic FASD studies to validate the findings of this study. Additionally, including trios (child FASD diagnosis, maternal and paternal genomes) in larger FASD studies would be insightful for many FASD-related analyses, such as PAE- and NDD-sensitizing variants. Lastly, this study was unable to have matched controls, nor is there an ideal reference population from which to draw control allelic frequencies. Currently, there is no public Canadian Indigenous Peoples database available to serve as a control for these kinds of genetic studies, although one is currently being assembled. While genetic differences are likely influencing the results of this study, the fact that individuals were segregated based on diagnosis potentially supports the methodology to use the 1000 Genomes Project dataset as a control cohort, despite the FASD cohort being heterogeneous. This suggests that ethnicity may not be the driving factor in the genetic differences found between the diagnostic groups and supports this study’s results to investigate a more diverse population in a similar methodology.

## 5. Conclusions

In this exploratory study, the goal was to investigate the variants found in the genes comprising the RA signaling network in 23 individuals with FASD. Novel FASD risk and resilience alleles in each of our three candidate gene lists were discovered: (1) RA and alcohol (ethanol) metabolism genes, (2) RA-interacting developmental pathway genes, and (3) the causative genes of rare NDDs and their direct signaling targets. Two previously known FASD protective variants were identified to be reduced in our cohort, and an additional 67 new potential FASD risk and resilience genetic variants were found. That these new variants were found in genes throughout the RA and alcohol metabolic pathways, in developmental pathways, and in known causative genes of NDDs that share co-morbidities with FASD suggests that there is a genetic vulnerability that sensitizes an individual to develop FASD in the presence of PAE. These results support the possibility that genetic biomarkers for the risk of FASD outcomes may help in the much earlier assessment of children with PAE and early signs of neurodevelopmental deficits. Surprisingly, the results of this study have shown that specific genetic variants of FASD risk and resilience may be able to be used to distinguish FASD outcomes in individuals of different FASD diagnoses and, moreover, possibly identify the risk of FASD outcomes in the larger majority of currently undiagnosed children with FASD. Ideally, this work will put us one step closer to developing a clinical genetic assessment tool to assess the risk of PAE teratogenesis and FASD diagnosis at a much younger age. This will allow for earlier intervention for FASD individuals that would significantly mitigate secondary disabilities associated with FASD and have a profound impact on the life course trajectory of individuals with this disorder and their families.

## Figures and Tables

**Figure 1 biomolecules-14-00569-f001:**
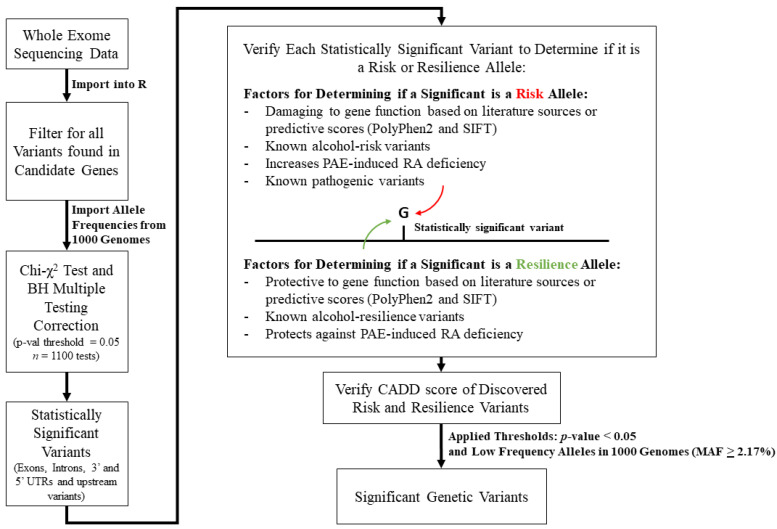
Flow chart of bioinformatic pipeline and determination factors considered for assigning risk and resilience alleles of genetic variants in the whole exome sequencing data of the FASD cohort. Once variants were imported into R, only variants found in our candidate genes were taken for analysis. This was followed by a chi-squared test using the 1000 Genomes Project allele frequencies as controls, which was followed by Benjamini–Hochberg multiple testing correction. Each variant was then sequentially determined to be an FASD risk or resilience allele based on defined criteria (literature sources and predictive scores) as seen in the Appendix A. The CADD scores of discovered risk and resilience alleles were then investigated. Lastly, thresholds were applied at *p*-values < 0.05 and for low-frequency alleles in 1000 Genomes, and only significant genetic variants passing thresholds are shown in the results tables.

**Figure 3 biomolecules-14-00569-f003:**
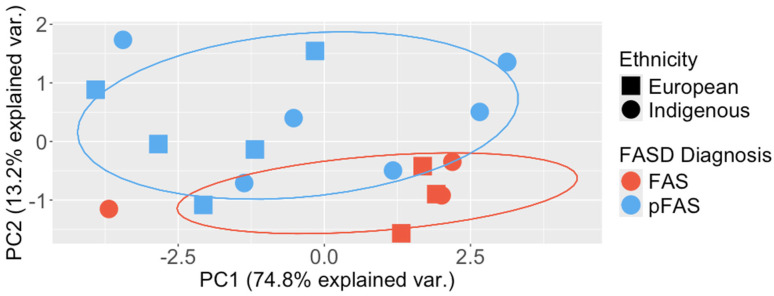
The FASD cohort clusters by diagnosis when using pathway−based polygenic risk scores. When examining pathway-based polygenic risk scores for each patient using only variants in genes common to individuals with a FAS and pFAS diagnosis, they were found to cluster by diagnosis, and not by ethnicity.

**Table 1 biomolecules-14-00569-t001:** Variant analysis results of retinoic acid and alcohol metabolism candidate genes.

rsID	Gene	Type of Mutation	Risk Allele	Biological Impact of Variant	FASD Risk Allele Frequency	1000 Gen. Risk Allele Frequency	*p*-Value
Known Alcohol Protective Alleles							
rs1229984 *	*ADH1B*	Missense	C	More alcohol consumption	0.96	0.82	1.70 × 10^−2^
rs1614972 *	*ADH1C*	Intron	C	T variant is associated with reduced alcohol dependence	0.93	0.48	5.61 × 10^−9^
rs2066702 *	*ADH1B*	Missense	G	A variant oxidizes ethanol fast	1.00	0.95	5.00 × 10^−3^
rs3762894 *	*ADH4*	Upstream	T	T allele is associated with less alcohol consumption	0.91	0.68	4.00 × 10^−3^
rs671 *	*ALDH2*	Missense	G	G allele is associated with reduced alcohol consump.	1.00	0.96	8.10 × 10^−4^
Known Alcohol Sensitizing Alleles							
rs2241894	*ADH1C*	Silent	C	C allele is associated with alcohol dependence	0.09	0.47	1.23 × 10^−6^
rs698 *	*ADH1C*	Missense	C	C variant is 1.5 to 2-fold less kinetically active	0.54	0.21	2.66 × 10^−6^
rs1612735	*ADH1C*	Intron	C	C allele is associated with alcohol dependence	0.52	0.21	1.77 × 10^−6^
rs971074	*ADH7*	Silent	T	T allele is associated with heroin addiction	0.22	0.12	3.00 × 10^−2^
rs2073478	*ALDH1B1*	Missense	G	TT genotype is protective against drinking	0.41	0.61	3.00 × 10^−2^
New Retinoic Acid Metabolizing Risk and Resilience Variants							
rs2298753 *	*ADH1C*	3′ UTR	C	VUS	0.72	0.94	7.28 × 10^−10^
rs10891338	*BCO2*	Missense	C	C allele probably has a damaging PolyPhen score	1.00	0.97	2.20 × 10^−4^
rs12263200 *	*CYP26C1*	Intron	G	T allele has a damaging SIFT prediction	0.80	0.88	1.00 × 10^−3^
rs58993699	*CYP26C1*	Intron	C	T allele has a damaging SIFT prediction	0.02	0.03	1.00 × 10^−3^
rs55962377	*DGAT1*	Missense	T	T allele has a damaging PolyPhen score	0.02	0.04	2.00 × 10^−3^
rs3829462	*LIPC*	Missense	A	A allele has a damaging PolyPhen score	0.65	0.94	3.30 × 10^−15^
rs2227277	*NCOR2*	Missense	C	T allele has damaging PolyPhen and SIFT scores	0.07	0.06	3.00 × 10^−2^
rs2134095 *	*RXRG*	Silent	A	A allele affects splicing: lower cholesterol and gest. diabetes	0.72	0.48	2.80 × 10^−3^
Variants in RA Metabolism Genes Associated with Congenital Malformations/Syndromes							
rs971756 *	*STRA6*	Missense	T	T variant damaging SIFT; Matthew–Wood syndrome	0.13	0.05	2.00 × 10^−3^
rs11857410 *	*STRA6*	Silent	A	A allele in microphthalmia; Matthew–Wood syndrome	0.13	0.06	5.00 × 10^−3^

* Validated by TaqMan genotyping.

**Table 2 biomolecules-14-00569-t002:** Variant analysis results of genes in retinoic acid-regulated pathways.

rsID	Gene	Type of Mutation	Risk Allele	Biological Impact of Variant	FASD Risk Allele Frequency	1000 Gen. Risk Allele Frequency	*p*-Value	Risk or Resilience
Variants Found in Genes Regulated Directly by Retinoic Acid Signaling								
rs2240308	*AXIN2* (3)	Missense	A	Cryptorchidism and congenital heart disease	0.65	0.34	3.64 × 10^−5^	Risk
rs1049007	*BMP2* (2)	Silent	A	Ossification defects (ligament)	0.47	0.25	1.70 × 10^−3^	Risk
rs45498702	*CAV1*	3′ UTR	T	T variant nominally associated with arterial fibrillation	0.07	0.04	4.00 × 10^−3^	Risk
rs1140475	*EGFR*	Silent	C	C variant assoc. with temporomandibular disorder	0.80	0.92	2.00 × 10^−3^	Resilience
rs2592595	*GLI2*	Silent	A	A variant found in a study of ectrodactyly patients	1.00	0.83	4.00 × 10^−3^	Risk
rs1044006	*NOTCH3*	Silent	C	C variant: cerebral infarction, mandibular prognathism	0.71	0.87	2.00 × 10^−3^	Resilience
rs2066836	*PTCH1*	Silent	A	A variant: non-syndromic cleft lip and palate	0.33	0.09	4.80 × 10^−8^	Risk
rs139884	*SOX10*	Silent	G	G variant: Waardenburg Type I and II syndromes	0.46	0.71	5.10 × 10^−4^	Resilience
rs1051886	*WNT10B*	Silent	A	A variant: hip bone mineral density	0.52	0.31	7.00 × 10^−3^	Risk
Variants Found in Genes Downstream of Retinoic Acid-Controlled Pathways								
rs8940	*CAV2*	Missense	G	G Variant has damaging PolyPhen and SIFT scores	0.33	0.14	5.50 × 10^−4^	Risk
rs1638630	*PTCHD3*	Missense	C	C Variant has damaging PolyPhen score	0.04	0.03	7.31 × 10^−5^	Resilience

(n) Number of significant variants in the same gene.

**Table 3 biomolecules-14-00569-t003:** Variant analysis of causative genes of rare neurodevelopmental disorders.

rsID	Gene	Type of Mutation	Risk Allele	Syndrome and Comorbidity	FASD Risk Allele Frequency	1000 Gen. Risk Allele Frequency	*p*-Value	Risk or Resilience
rs55975541	*CDC42BPG* *	Missense	A	Aarskog	0.11	0.03	3.66 × 10^−6^	Risk
rs75659311	*FGD1* †	Missense	A	Aarskog	0.04	0.02	4.48 × 10^−5^	Risk
rs1047057	*FGFR2* *	Silent	G	Apert	0.65	0.41	5.00 × 10^−3^	Risk
rs2229989	*SOX9* *	Silent	T	Campomelic Dysplasia	0.24	0.14	4.00 × 10^−2^	Risk
rs80068543	*ARVCF* *	Missense	T	22q11.2 Deletion	0.02	0.04	2.00 × 10^−3^	Resilience
rs712952	*CLTCL1* † (2)	Missense	A	22q11.2 Deletion	0.11	0.05	3.30 × 10^−3^	Risk
rs4819756	*PRODH*	Missense	A	22q11.2 Deletion	0.54	0.22	5.20 × 10^−7^	Risk
rs2277838	*P2RX6*	Missense	A	22q11.2 Deletion	0.13	0.08	3.50 × 10^−2^	Resilience
rs2227902	*REST* *	Missense	T	22q11.2 Deletion	0.09	0.07	3.60 × 10^−2^	Risk
rs874100	*SCARF2* (3)	Missense	C	22q11.2 Deletion	0.04	0.24	5.00 × 10^−3^	Resilience
rs5746826	*TBX1* * (2)	3′ UTR	T	22q11.2 Deletion	0.30	0.59	4.00 × 10^−4^	Risk
rs3747052	*TSSK2*	Missense	G	22q11.2 Deletion	0.02	0.04	2.90 × 10^−3^	Resilience
rs2871053	*ZDHHC8*	Upstream	C	22q11.2 Deletion	0.07	0.05	1.40 × 10^−2^	Risk
rs2230808	*ABCA1*	Missense	T	Smith–Lemli–Opitz	0.76	0.58	1.00 × 10^−2^	Risk
rs10932816	*OBSL1*	Silent	G	Smith–Lemli–Opitz	1.00	0.96	8.90 × 10^−4^	Risk

* Directly regulated by retinoic acid; † upstream or downstream of retinoic acid-controlled pathways; and (n) number of significant variants in the same gene.

## Data Availability

The 1000 Genomes Project frequencies employed in this study can be found on NCBI (https://www.ncbi.nlm.nih.gov/snp/, accessed on 2 March 2022). The original data and contributions presented in this study are included in the article/Appendix A; further inquiries can be directed to the corresponding author.

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
