# Peer review of "Risk and Resilience Variants in the Retinoic Acid Metabolic and Developmental Pathways Associated with Risk of FASD Outcomes"

_biomolecules, 2024, doi:10.3390/biom14050569_

Round 1
Reviewer 1 Report
Comments and Suggestions for Authors
biomolecules-2973239
The manuscript by McKay et al., entitled 'Risk and Resilience Variants in the Retinoic Acid Metabolic and Developmental Pathways Associated with Risk of FASD Outcomes,' addresses a highly interesting topic regarding the penetrance (and possibly the prevalence) of FASD. Although thus far no monogenetic variation has been identified, the authors hypothesized that for most, common genetic variation may contribute to susceptibility to developing FASD upon in utero exposure to alcohol. The manuscript is generally well-written regarding grammar and spelling. However, the structure of the manuscript could be improved. Tables should be ordered within the section they belong, and, more importantly, the general text could be shortened. While I find this manuscript potentially suitable for publication, there are several major points that need to be addressed.
1. The authors clearly claim that genes previously associated with alcohol dependency may also play a functional role in the manifestation of FASD. This claim cannot be made, since these alleles originate from the mother, or even from the father, since alcohol dependency is typically also a household problem. A substantial number of genetic features should therefore be placed outside the context of a functional relationship with FASD, but rather be positioned as a control set, proving that certain enrichments in this small cohort can be detected.
2. In Table 4, the size of the sample is simply too small to stratify for pFAS, FAS, or ARND. The authors should therefore downscale any claims made here and focus solely on FASD in general. For instance, allele frequencies based on only 5 individuals with ARND are highly unreliable. A report on general FASD is therefore sufficient, and this study should be presented as explorative, earlier than in conclusion.
3. The absence of an ethnicity-matched control sample is a drawback in this study. Notably, alcohol dependency, although perhaps due to socio-economic reasons rather than genetics, is a strong confounder and can only be addressed by introducing a control sample. In my opinion, there is no valid reason to omit this highly relevant aspect within a genetic study. In this context, the 1000 Genomes dataset is certainly not sufficient, as this cohort may not include the ethnic background of the FASD sample.
4. Within genetic studies, whether to use sibling pairs or not, is strongly dependent on the study design. In this study, siblings are obviously a confounding factor. The present study has not addressed this issue, which may have resulted in confounding and an increased allele frequency of certain alleles. In that context, siblings, i.e., one of the samples, should have been removed from the sample.
5. In particular, in this study where genetic variants/genes previously associated or linked with NDD are examined, clinical characterization is of utmost importance. As the authors indicate, FASD shares phenotypic characteristics, emphasizing the importance of correct diagnosis. While the method section briefly refers to the clinical diagnosis methodology, it is virtually absent in the text. The gold standard here is, for example, the 4-digit score. Furthermore, since this FASD characterization is particularly important, such diagnosis should have been independently performed by two clinicians to exclude any other NDD. If the referred guidelines actually include such an approach, the authors should briefly describe this procedure.
6. The authors describe that WES analysis was performed in all FASD cases. In order to include or exclude any de novo variant, trio analysis should have been performed. However, this limitation is not included in the manuscript.
7. The issue of ethnicity, although reported in PCA using the candidate SNPs, should have been included for all variations. Moreover, the ethnicity overlap of the 1000 Genomes would have also been informative, especially for candidate SNPs. Probably the outcome of this would contribute to convincing readers.
Minor:
Line 146, a systemic review requires an in depth description and otherwise authors should rephrase.
Tables: P values should presented consistently as scientific number.
Author Response
Thank you very much for taking the time to review our paper. Please see attachment with responses to your comments.

Reviewer 2 Report
Comments and Suggestions for Authors
The current manuscript entitled “Risk and Resilience Variants in the Retinoic Acid Metabolic and Developmental Pathways Associate with Risk of FASD Outcomes” by Leo McKay et al. is well written and presented. The findings will be a fantastic contribution in the FASD research. However, the following minor modifications need to be addressed before making any final decision.
1# In page 3, line number 113 the author should elaborate pFAS and then use the abbreviated form throughout the manuscript.
2# In the Method section the author needs to add company names of SOLiD A1; P1 adaptors; ABI SOLiD 5500x. Furthermore, need to cite a reference for Agilent SureSelect Protocol. Authors are suggested to use the international format of company name (name, city, country) throughout the manuscript.
3# In the Result section the author cited a huge number of publications. I am wondering!! The result section will represent original results they have. And discuss those findings with others in the discussion section with different dimensions. The authors are suggested to merge the Result and Discussion section if the journal permits, Or they need to rewrite the Result section removing all the citation they included.
4# It is preferable to write a manuscript using the style of third gender format. The authors are suggested to replace all we/our words from their manuscript and rewrite it accordingly.
Author Response
Thank you for taking the time to review our paper. Please see the attachment for responses to your comments.

Round 2
Reviewer 1 Report
Comments and Suggestions for Authors
The authors have sufficiently addressed all of my previous concerns.